# An Optimized Single Nucleotide Polymorphism-Based Detection Method Suggests That Allelic Variants in the 3’ Untranslated Region of *RRAS2* Correlate with Treatment Response in Chronic Lymphocytic Leukemia Patients

**DOI:** 10.3390/cancers15030644

**Published:** 2023-01-19

**Authors:** Alejandro Hortal, Marta Lacuna, Claudia Cifuentes, Miguel Alcoceba, Xosé R. Bustelo, Marcos González, Balbino Alarcón

**Affiliations:** 1Immune System Development and Function Program, Centro Biología Molecular Severo Ochoa, Consejo Superior de Investigaciones Científicas (CSIC)-Universidad Autónoma de Madrid, 28049 Madrid, Spain; 2Departamento de Hematología, Hospital Universitario de Salamanca (HUS-IBSAL), 37007 Salamanca, Spain; 3Centro de Investigación del Cáncer, Instituto de Biología Molecular y Celular del Cáncer, Centro de Investigación Biomédica en Red de Cáncer, Centro Biología Molecular Severo Ochoa, Consejo Superior de Investigaciones Científicas (CSIC), Universidad de Salamanca, 37007 Salamanca, Spain

**Keywords:** *RRAS2*, *RAS*, GTPases, SNP, polymerase chain reaction, PCR, chronic lymphocytic leukemia, CLL, cancer progression, cancer prognosis, ibrutinib, Bruton kinase, cancer treatment

## Abstract

**Simple Summary:**

Overexpression of unmutated *RRAS2* could be behind the development of chronic lymphocytic leukemia (CLL). This is, in part, supported by the presence of a single-nucleotide polymorphism (SNP) in the 3’ untranslated region of the *RRAS2* mRNA that is linked to higher *RRAS2* overexpression and worse prognosis. Determining the nucleotide composition (G or C) at the SNP position might be of prognostic value and helpful in orienting the choice for therapies. To make SNP analysis more feasible, we show here the implementation of a new PCR method that allows the characterization of the SNP allele composition with a little amount of genomic DNA. As an example, we show in a small subset of CLL patients that treatment with the drug Ibrutinib results in a stronger reduction in circulating leukemic cells if the patients bear at least one RRAS2 allele with a C at the SNP position.

**Abstract:**

Unlike classical *RAS* genes, oncogenic mutations on *RRAS2* are seldomly found in human cancer. By contrast, *RRAS2* is frequently found overexpressed in a number of human tumors, including B and T cell lymphomas, breast, gastric, head and neck cancers. In this regard, we have recently shown that overexpression of wild-type *RRAS2* drives the formation of both chronic lymphocytic leukemia (CLL) and breast cancer in mice. In support for the relevance of overexpression of wild type *RRAS2* in human cancer, we have found that *RRAS2* expression is influenced by the presence of a specific single nucleotide polymorphism (SNP) located in the 3’-untranslated region (UTR) of the *RRAS2* mRNA. Perhaps more importantly, the presence of the alternate C, rather than the G allele, at the *RRAS2* SNP designated as rs8570 is also associated with worse patient prognosis in CLL. This indicates that the detection of this SNP allelic variants can be informative to predict *RRAS2* expression levels and disease long-term evolution in patients. Here, we describe a polymerase chain reaction (PCR)-based method that facilitates the rapid and easy determination of G and C allelic variants of the SNP. Using this approach, we confirm that the C allelic variant is associated with higher expression levels of *RRAS2* transcripts and poor patient prognosis. However, we have also found that expression of the C allelic variants correlates with better response to ibrutinib, a Bruton kinase inhibitor commonly used in CLL treatments. This suggests that this method for detecting the *RRAS2* rs8570 SNP might be a useful as a tool to predict both patient prognosis and response to targeted therapy in CLL.

## 1. Introduction

B-Cell Chronic Lymphocytic Leukemia (CLL) is the most common cause of leukemia in the Western world with a prevalence of 2:1 in male and female patients, respectively [1,2]. CLL is a disease that mainly affects the elderly population, with a median age of diagnosis of 72 years [3,4]. Indeed, in patients older than 80 years of age, the incidence of CLL rises from 4.2 in the general population to 30 per 100.000 per year [3]. The diagnosis of CLL is denoted by ≥5 × 10^9^/L monoclonal B cells in the peripheral blood with a characteristic immunophenotype (CD19+/CD5+/CD23+/CD200+) [1]. Current data show that the 5-year survival rate of CLL is 87.9% [5], a significant improvement from the late years of the 2000s, when this figure was at 77.3% [6].

In cases that do not present with lymphadenopathy or organomegaly, a B lymphocyte count below the given threshold is termed as monoclonal B lymphocytosis (MBL). MBL has been found to progress to CLL at a rate of 1–2% per year [4]. CLL has a very heterogeneous clinical course that ranges from patients with a stable form of the disease that does not require treatment (watch and wait patients), while others develop an aggressive form of CLL. Some clinical and biological parameters (male sex, age > 60 years, advanced clinical stage, higher absolute lymphocyte count, high levels of serum β2-microglobulin and LDH) together with molecular and cytogenetic characteristics (presence of del (17p) and/or *TP53* mutations, presence of del(11q), and unmutated *IGHV* gene, have been associated to poor prognosis [4,7].

BCR signaling is significantly involved in CLL, since its repeated stimulation upregulates CD5 expression, a marker normally found in T cells that is also a phenotypic marker for CLL [8]. The relevance of BCR signaling is also highlighted by some of the current treatment options, which focus on inhibiting BCR associated kinases. Such is the case for ibrutinib, an inhibitor of Bruton’s tyrosine kinase (BTK), fosfamatinib, an inhibitor of spleen tyrosine kinase (SYK), or idelalisib, which targets the hematopoietically expressed PI3Kδ [9]. The selection of a specific treatment is influenced by the genomic landscape of the patient, mainly the mutational status of the *IGHV* genes and del(17p) and/or mutation at *TP53*. These options include preferentially the aforementioned inhibitors, as well as the combination therapy FCR (fludarabine, cyclophosphamide and rituximab) and the anti-apoptotic factor BCL2 inhibitor venetoclax [3].

RAS proteins are a family of small guanosine triphosphate hydrolases (GTPases) that include well known oncogenic players such as K-RAS, H-RAS and N-RAS. The RAS-related subfamily of RAS proteins (R-RAS) share approximately 60% amino acid identity with their classic counterparts as well as associated proteins that regulate their activation-inactivation cycles, i.e., guanine nucleotide exchange factors (GEFs) and GTPase-activating proteins (GAPs). Early studies from the 1990s showed that oncogenic mutations in R-RAS2 (G23V, Q72L) analogous to those found in classical RAS proteins (G12V, Q61L) have similar or even higher transformation potential [10,11]. *RRAS2* has been described to play an essential role in immune system development and homeostasis. It binds both the B and T cell receptors (BCR and TCR, respectively) and mediates the generation of tonic survival signals from these receptors [12]. Specifically in B cells, R-RAS2 is essential for the proper formation of a correct germinal center reaction by means of the regulation of B cell metabolism [13].

*RRAS2* has been found bearing activating mutations in patients with the rare condition Noonan Syndrome (NS), characterized by craniofacial defects, short stature, congenital heart disease, predisposition to malignancies and variable cognitive impairment [14,15]. *RRAS2* has also been found to be overexpressed in different types of human malignancies: skin cancers [16], oral cancers [17] or esophageal tumors [18], although a causal relationship has not yet been established. 

We have recently shown that overexpression of *RRAS2* in its wild type form, but not its oncogenic mutant Q72L, drives the development of CLL in human patients and also in mouse models of human *RRAS2* overexpression in B cells [19]. The overexpression of this small GTPase is associated to more lymphocytosis and to a higher percentage of CD19+CD5+ malignant cells in the blood of the patients. Moreover, it is increased in patients with full-blown CLL compared to patients with the pre-malignant stage monoclonal B-cell lymphocytosis (MBL) and it is increasingly overexpressed in male versus female patients [19]. This is consistent with data showing that CLL affects males approximately twice as much as females and have a poorer prognosis [2]. Apart from the overexpression, we described a single nucleotide polymorphism (SNP) (rs8570) that involves the change in the canonical G nucleotide at position 124 of *RRAS2* 3’UTR for a C. The presence of a C allele in homozygosity or heterozygosity at this position has significantly been associated with different parameters of worse prognosis: higher *RRAS2* expression, higher lymphocytosis, higher numbers of leukemic CD19+CD5+ cells, lower platelet count, higher percentage of full-blown CLL at the time of diagnosis compared to MBL, more chromosomal alterations detected by fluorescence in situ hybridization (FISH), more patients with un-mutated *IGHV* genes and it was more frequent in male than in female patients. In this report, we propose a new technique to optimize the characterization of this rs8570 SNP that requires a considerably lower amount of sample and is more time-efficient. This will allow the analysis of rs8570 in multiple malignancies and favor its potential use as a new predictive/prognostic factor of disease severity and response to targeted therapies.

## 2. Materials and Methods

### 2.1. Human Cells

Human blood samples were obtained from volunteer CLL patients from the Hematology Unit of the Salamanca University Hospital after providing written informed consent, with authorization number PI 2019 03217. Fresh human peripheral blood mononuclear cells (PBMCs) were obtained by density centrifugation in a Lymphoprep™ (StemCell technologies, Vancouver, BC, Canada) gradient of whole blood for flow cytometry, RT-qPCR analysis and gDNA extraction.

### 2.2. Flow Cytometry

Human single-cell suspensions were incubated for 15 min with Ghost Dye™ 540 (Tonbo, San Diego, CA, USA) in PBS to label and discard dead cells from analysis. Cells were washed with PBS + 2% FBS before being incubated with fluorescently labelled antibodies for 30 min at 4 °C. Afterwards, cells were washed in PBS + 2% FBS and data were collected on a FACS Canto II (Becton Dickinson, Franklin Lakes, NJ, USA) cytometer. A minimum of 50,000 and a maximum of 200,000 events was acquired in every measurement. Analyses were performed using FlowJo software (TreeStar, Ashland, OR, USA).

### 2.3. gDNA Extraction

A total of 10^6^ cells were harvested per sample. A volume of 500 μL of lysis buffer (Tris-HCl pH8 50 mM, NaCl 200 mM, EDTA 10 mM, SDS 1% and fresh proteinase K 0.2 mg/mL) were added to each sample and incubated overnight (ON) at 55 °C. The next day, gDNA was purified using phenol chloroform and resuspended in 100 μL of 10 mM Tris-HCl pH 8.5 depending on the pellet size obtained after the last centrifugation step.

### 2.4. Sequencing Strategy for Patients’ Samples

Ten million cells per patient sample were used for RNA extraction from PBMCs of CLL patients. RNA was isolated using the RNAeasy Plus Mini Kit (Qiagen, Hilden, Germany). cDNA was synthesized with SuperScript III (Invitrogen) using Oligo-dT primers. cDNA was used to sequence the 3’UTR region of *RRAS2*. Specific oligonucleotides were used to detect the presence of C or G allele by qPCR using the patients’ cDNA as template. Quantitative real-time PCR was performed in triplicate using 100 ng cDNA and the reverse transcription reaction with SYBR Green PCR Master Mix and gene-specific primers in an ABI 7300 Real Time PCR System. These oligonucleotides were designed as described in [20]. Their sequences can be provided upon request. As a loading reference, specific oligonucleotides aligning to constitutive *RRAS2* exons 3 and 4 were used (Forward: GCA GGA CAA GAA GAG TTT GGA; Reverse: TCA TTG GGA ACT CAT CAC GA). Obtained cycle threshold (Ct) values were used to calculate 124G and 124C mRNA levels relative to exons 3 and 4 expression using the 2^−ΔΔCt^ method. All oligonucleotides are indicated by their 5′–3′ orientation. Alternatively, a nested PCR was performed for rs8570 sequencing, using the approach explained in [19].

### 2.5. RRAS2 Expression Measurement

A set of primers that expand constitutive exons 3 and 4 was used to measure mRNA expression of *RRAS2* in patient PBMCs (Forward: GCA GGA CAA GAA GAG TTT GGA; Reverse: TCA TTG GGA ACT CAT CAC GA). Obtained cycle threshold (Ct) values were used to calculate mRNA levels relative to 18S rRNA expression using the 2^−ΔΔCt^ method. Outliers for *RRAS2* expression were identified using ROUT model at Q = 0.1% to remove definite outliers from analysis.

### 2.6. Fluorescent Probes Method to Sequence Patients’ Samples

Fluorescent dual-labelled probe technology, pre-developed by Applied Biosystems (Foster City, CA, USA) and custom made to target rs8570 SNP was used to optimize its characterization. Briefly, this technology requires a mix of two PCR primers that anneal upstream and downstream of the analyzed SNP and two allele-specific TaqMan^®^ MGB (minor groove binder) probes, labelled with FAM™ and VIC™ fluorescent dyes. TaqMan^®^ MGB probes consist of target specific oligonucleotides with a reporter VIC™ dye at the 5’ end of the probe for allele 1 (124G) or a reporter FAM™ dye for allele 2 (124C). Both probes present a non-fluorescent quencher (NFQ) dye at the 3′ end of the probe. AmpliTaq Gold™ DNA polymerase used in this assay only cleaves probes with 100% complementarity to the sample DNA, liberating the specific fluorescent probes and emitting fluorescence that can be detected due to the loss of the NFQ. Reactions were performed using an initial cycle of 10 min at 95 °C for polymerase activation, followed by 40 cycles of 15 s at 95 °C and 1 min at 60 °C. rs8570 allele distribution was assessed by plotting the relative fluorescent units (RFU) of both fluorophores in a scatter-plot representation. An alternate form representation and classification into the three genotypes (GG, GC and CC) was calculated for each individual point in the VIC RFU–FAM RFU plot, by plotting the distance and angle relative to the origin (0,0). Firstly, using Pythagoras theorem we calculate the distance of each point relative to the origin:(1)$$Distance=\sqrt{{\left(\Delta \;VIC\;RFU\right)}^{2}+{\left(\Delta \;FAM\;RFU\right)}^{2}}$$

This value corresponds to the y-axis value in the new plot. Secondly, the x-axis value corresponds to the angle (*θ*) calculated for each point according the tangent from the VIC RFU–FAM RFU plot as follows:(2)$$Angle\;\left(\theta \right)=\tan^{-1}\left(\frac{\Delta \;VIC\;RFU}{\Delta \;FAM\;RFU}\right)$$

## 3. Statistical Analysis

Statistical parameters including the exact value of *n*, the mean +/− S.D. or S.E.M. are described in the Figures and Figure legends. Two-tailed Student’s t test with Welch’s correction was used as indicated to assess the significance of mean differences. A Chi-square test was used to assess if the allele distribution fell under a Hardy–Weinberg equilibrium. All data were analyzed using the GraphPad Prism 7 software.

## 4. Results

### 4.1. A Novel qPCR Method Greatly Improves Sensitivity and Time Efficiency of rs8570 SNP Characterization

For SNP allele composition, we previously [19] employed a qPCR with specific oligonucleotides for the detection of either G or C at rs8570, designed as described in [20], using the CLL patients’ PBMCs cDNA as template. In parallel, we confirmed the qPCR results by Sanger sequencing of a nested PCR product also obtained from the patients’ cDNA. Nevertheless, both approaches are sample- and time-consuming. Due to this, the development of efficient and reliable high throughput genetic screening methodologies that minimize project timelines is required to optimize the process of rs8570 characterization. With this purpose, we have approached the utilization of a custom-made TaqMan^®^ SNP Genotyping Assay (Applied Biosystems; Foster City, CA, USA). This method increases the processing ability of the qPCR by 200%. The previous method needed three parallel qPCRS, preparing three sets of triplicates for each sample (a set each containing primers hybridising to 124G, 124C and RRAS2 constitutive exons 3 and 4, respectively). Using the 2^−ΔΔCt^ method, we could identify three distinct populations (GG, GC and CC) corresponding to the SNP allele frequency distribution [19]. With this experimental layout it was only possible to characterize 42 samples (in triplicate) or 32 samples (in quadruplicate) per 384-well plate (one complete qPCR plate). Conversely, the fluorescent dual-labelled PCR enables the characterization of 128 samples with triplicate readings or 96 samples with quadruplicate readings, from a single 384-well plate. On top of this, the method we propose is sensitive to final DNA sample concentrations that exceed 0.2 ng/µL and according to the TaqMan^®^ SNP Assays protocol the recommended total cDNA/gDNA well amount should lie in the range of 1–20 ng. Comparably, our previous qPCR required cDNA to be added for a final amount of 100 ng per well. Therefore, the new method greatly reduces the amount of starting material required and achieves accurate characterization whilst minimizing contamination with reaction inhibitors.

In order to validate the new fluorescent probe method for rs8570 SNP sequencing, we re-genotyped all 178 patient samples characterized in [19] with 51 additional samples using both the older (double qPCR) and the new (single qPCR) approaches. The older method showed a disproportion of GG, GC and CC genotypes that were not at a Hardy–Weinberg equilibrium (Figure 1a). Such disequilibrium in allele distribution was mostly due to an observed enrichment in CC homozygotes at expenses of GC heterozygotes. The results generated using the single qPCR with the TaqMan^®^ SNP Genotyping Assay of the expanded cohort of patient’s samples showed that the population was at a Hardy–Weinberg equilibrium (Figure 1b, χ^2^ experimental = 0.08 < χ^2^ theoretical = 3.84), losing the higher-than-expected frequency of CC homozygotes and lower frequency of GC heterozygotes that we found with our prior cohort of untreated patients [19]. Upon representation of the RFUs from each of the two specific fluorescent probes for each allele of all patients, three clear populations could be established, illustrating the three possible allele compositions (Figure 1c). As in our previous study, expression of the C allele in homozygosis is significantly associated with higher expression of mRNA in leukemic cells (4.4-fold mean overexpression over that of healthy controls for GG, 4.9-fold for GC and 6.2-fold for CC; Figure 1d).

In pursuance of escalating the applicability of the new qPCR, it is of remarkable importance to be able to analyze directly genomic DNA (gDNA) instead of cDNA. gDNA can be easily obtained and, with as little as a drop of blood, it can be extracted and analyzed. cDNA, meanwhile, requires the prior purification of RNA, a process that needs more starting material and that renders a final product considerably less stable. Moreover, conversion to cDNA additionally requires the subsequent reverse transcriptase PCR reaction (RT-PCR). In 39 of the new 51 patients’ samples, we were able to test the new single qPCR method using gDNA as template. As shown in Figure 2, a scatter plot displaying the VIC RFU vs. FAM RFU values (Figure 2a) and a scatter plot representing the distance and angle of each point relative to the origin (Figure 2b) allowed clear grouped separation of the three allelic populations with a single reaction carried out with less than 20 ng template DNA.

### 4.2. Treated CLL Patients with at Least One C Allele at rs8570 Show a Better Response to Ibrutinib

As a proof-of-concept of the new genotyping qPCR strategy, we analyzed some clinical and RRAS2-expression data within the new cohort of 51 CLL patients. Among those there were pre- and post-treatment samples of nine patients on Ibrutinib therapy. Analysis of the pre- and post-treatment samples showed a general reduction in total leukocytes and lymphocytes, but not of platelets (Figure 3a–c). However, none of those differences reached statistical significance, likely due to the limited number of samples. In contrast to the Ibrutinib-treated group, another subset of patients for which we had repeated blood extractions, showed a non-significant increase in the number of total leukocytes and total lymphocytes in blood (Figure 3d–f). Interestingly, the comparison of *RRAS2* mRNA expression established by RT-qPCR of the 178 samples from non-treated CLL patients with that of the 14 Ibrutinib-treated patients of the new 51 patient cohort, showed a significant reduction in *RRAS2* expression in treated patients (Figure 4a). Such effect exceeded that of Ibrutinib therapy on total leukocyte and total lymphocyte numbers in blood (Figure 4b,c), thus ruling out that reduction in *RRAS2* expression was simply due to an effect on the number of leukemic cells. Those results rather suggest that Ibrutinib treatment somehow impacts the transcription or stability of the mRNA for *RRAS2*.

We previously found that expression of the C allele in one or two doses (GC+CC) was associated to higher lymphocytosis in untreated CLL patients [19]. To determine if in the new 51 patient cohort there was an association between the SNP allele composition and leukocytosis and lymphocytosis, we analyzed those parameters within the total number of 26 Ibrutinib-treated patients classified as GG homozygous or GC+CC (Figure 5a–c). Unexpectedly, patients with a GC or CC genotype on Ibrutinib treatment presented fewer total leukocytes and fewer total lymphocytes than patients with GG genotype (Figure 5a,b). Such a difference was not detected in the group of patients that were not on therapy (Figure 5d–f). Such differences in the response to Ibrutinib treatment in patients with GG vs. GC+CC genotype could not be explained by differences in age, sex or IGHV mutational status (Appendix A).

## 5. Discussion

The data presented in this report illustrate a faster, more efficient approach that requires considerably less amount of test sample for the genotyping of CLL samples for SNP rs8570. This system renders a 200% sample processing increase per 384-well used for each qPCR compared to our traditional qPCR strategy. Thus, it allows for a rapid escalation capability and can successfully be used for genotyping using small amounts of genomic DNA. The use of the novel single-qPCR system to genotype a cohort of 229 CLL samples (178 previously analyzed plus 51 additional ones) has resulted in loss of the previously found disequilibrium in allele composition with an overrepresentation of CC homozygotes [19]. Such increase beyond random distribution is nonetheless still found in the extended patient cohort (*n* = 229 patients) using the old double-qPCR method. The analysis of RRAS2 mRNA expression confirms the previous finding that the C allele is associated with higher mRNA expression, resulting statistically significant for leukemias with the C allele in homozygosis.

Unlike our previous cohort of 178 samples of CLL patients that were not needing treatment at the time of sampling, the extended cohort includes a total of 26 samples of CLL patients under treatment with Ibrutinib (median time of treatment = 13 months). Interestingly, the analysis of those patients classified according to the presence or not of the C allele, shows lower leukocytosis and lymphocytosis in Ibrutinib-treated patients if they have one or two C alleles. As previously reported [19], the allele C is associated to a worse prognosis in untreated patients, put on a watch and wait regime while the disease is not progressing. On the other hand, expression of the C allele seems to be also related to a better response to Ibrutinib treatment. Since expression of the C allele is linked to higher RRAS2 mRNA expression, those data would suggest that highest *RRAS2* expression sensitizes the leukemic cells to Ibrutinib. This conclusion needs however to be taken with caution given the low number of Ibrutinib-treated patients analyzed so far. We do not know the reasons for such increased sensitivity to Ibrutinib but it seems clear from our data that Ibrutinib treatment reduces *RRAS2* expression in CLL patients. Ibrutinib is an inhibitor of the tyrosine kinase Btk that signals downstream of the B-cell Receptor (BCR) [21,22]. R-RAS2 protein is, on the other hand, directly associated to the BCR in leukemic cells. We could, therefore, propose that at least part of the anti-leukemic effects of Ibrutinib are mediated by the downregulation of *RRAS2* expression, a process that might be controlled by the BCR and Btk.

SNP rs8570 was already identified to be associated with risk of cutaneous melanoma [23]. Our results described here need to be confirmed with a larger cohort of patients and it should also be analyzed if the type of treatment, combined with SNP rs8570 characterization, can be related to the extent of response to treatment and with the potential to relapse. With the qPCR method reported here, it will be quite easy to determine SNP rs8570 genotype in CLL patients as a new marker of prognosis and also of response to Ibrutinib therapy and perhaps other therapies. Our future works will also be oriented to study if SNP rs8570 can also be of value as a predictive/prognostic marker in other malignancies where *RRAS2* has been described to be involved, such as breast cancer [24] (Cifuentes et al., submitted), oral cancers [17], esophageal tumors [18] and lymphomas [12], as well as in other cancers in which the role of *RRAS2* overexpression is yet to be studied.

## 6. Conclusions

The single qPCR method is an efficient system to determine the allelic composition at the SNP rs8570 position in the 3’-untranslated region of *RRAS2* and seeks to correlate prognosis and response to therapies.

## Figures and Tables

**Figure 1 cancers-15-00644-f001:**
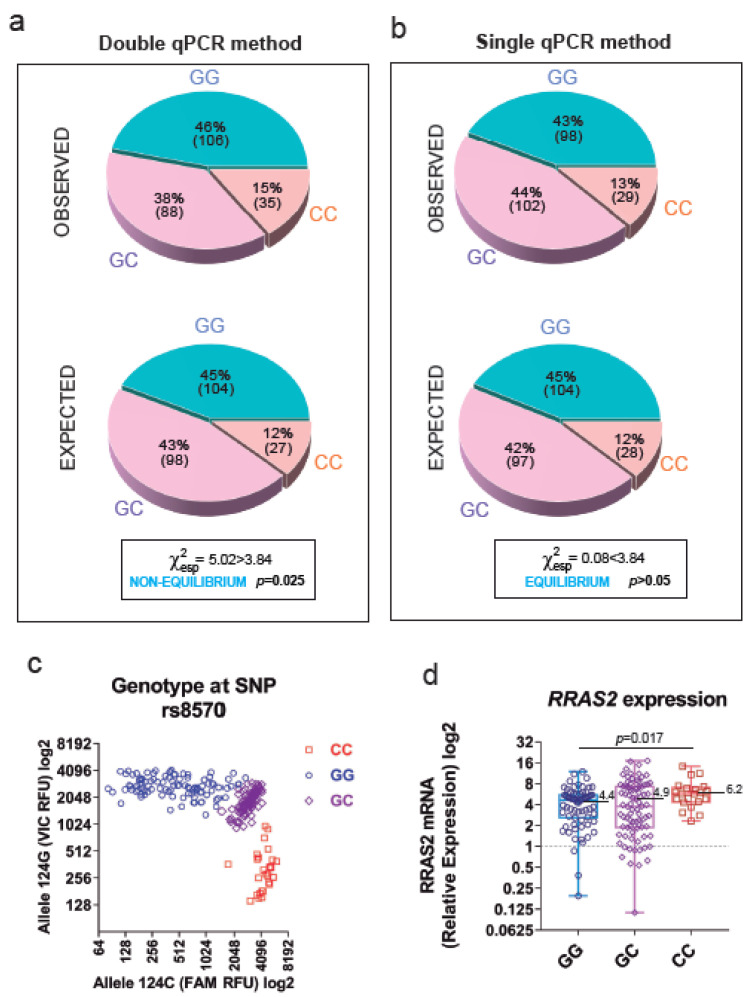
The new genotyping qPCR strategy for SNP rs8570 shows the G and C alleles to be at equilibrium. (**a**) Pie chart representation of the Observed and Expected (according to Hardy–Weinberg) allele distribution of SNP rs8570 in our expanded (*n* = 229) patient cohort according to our older qPCR method based on two separate PCRs. (**b**) Pie chart representation showing the Observed and Expected allele distribution after analysis using the new single-PCR method. Allele distribution was considered at equilibrium if the Chi-square test yielded a χ2 < 3.84. (**c**) Scatter plot representation of the RFUs of the VIC™ (124G allele) and FAM™ (124C) fluorophores in all the expanded patient cohort (*n* = 229). (**d**) *RRAS2* mRNA expression levels in CLL patients (*n* = 174) classified according to their GG, GC or CC genotype. Expression values are normalized to that of the mean of 10 healthy blood donors (set as 1). Statistical significance was assessed using a two-tailed unpaired t-test with Welch’s correction. GG, blue circles; GC, purple diamonds; CC, red squares.

**Figure 2 cancers-15-00644-f002:**
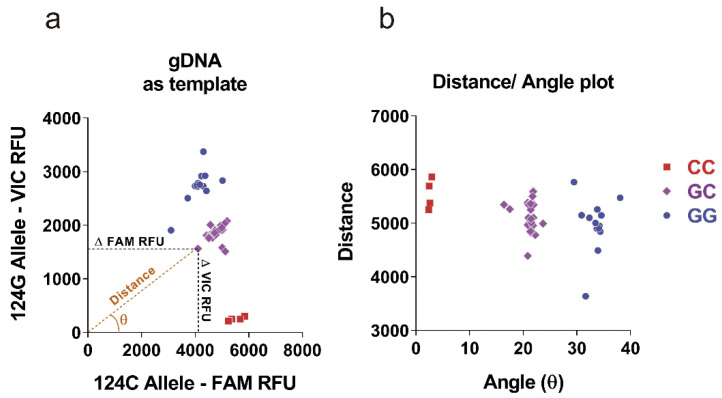
gDNA can be used as template for rs8570 genotyping. (**a**) Scatter plot representation of the RFUs of the VIC™ (124G allele) and FAM™ (124C) fluorophores using gDNA as template in a representative group of 39 patients. (**b**) For each value of the scatter plot in (**a**) a new value of the distance to the origin (0,0) and a value of the angle with respect to the x-axis were calculated and plotted in another scatter plot that allowed a better resolution of the three genotypes (GG, GC and CC).

**Figure 3 cancers-15-00644-f003:**
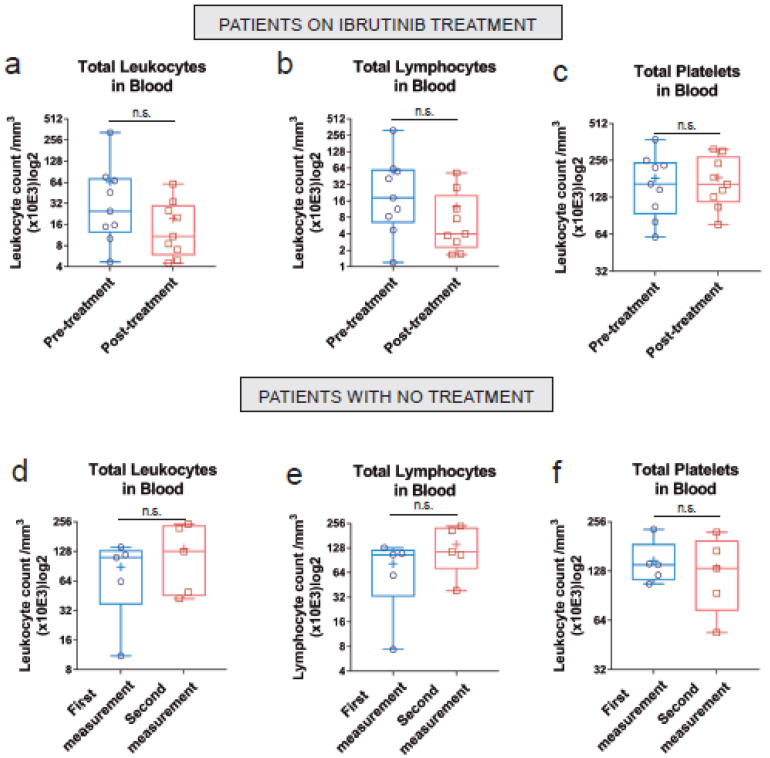
Non-significant reduction of leukocytosis after chemotherapy in our small cohort of Ibrutinib-treated. (**a**–**c**) Box and whiskers plots showing all points, median and mean values of the total leukocyte, lymphocyte and platelet count, respectively, in the blood of treated patients. The left column corresponds to the samples obtained prior to treatment initiation and the right column to that after its initiation (*n* = 9). Time between pre- and post-treatment timepoints: mean and median = 18 months. (**d**–**f**) Box and whiskers plots showing all points, median and mean values of the total leukocyte, lymphocyte and platelet count, respectively, in the blood of non-treated patients for whom more than one sample at different time points was available (*n* = 5) Time interval between the two blood draws: mean = 14 months, median = 12 months. n.s: not significant (unpaired two-tailed t-test with Welch’s correction).

**Figure 4 cancers-15-00644-f004:**
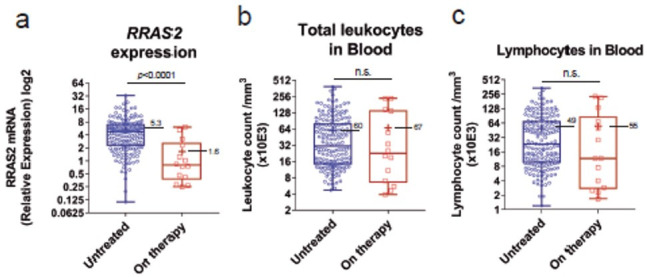
*RRAS2* mRNA expression is significantly reduced in Ibrutinib-treated CLL patients. (**a**) Box and whiskers plots showing all points (*n* = 193), median and mean values of the RRAS2 mRNA relative expression of untreated patients (left column; *n* = 179) and Ibrutinib-treated patients (right column; *n* = 14). (**b**,**c**) Box and whiskers plots of the total leukocyte and lymphocyte counts respectively, in the blood of untreated (left column) and Ibrutinib-treated (right column) patients. Statistical significance was assessed using a two-tailed unpaired t-test with Welch’s correction. n.s: not significant.

**Figure 5 cancers-15-00644-f005:**
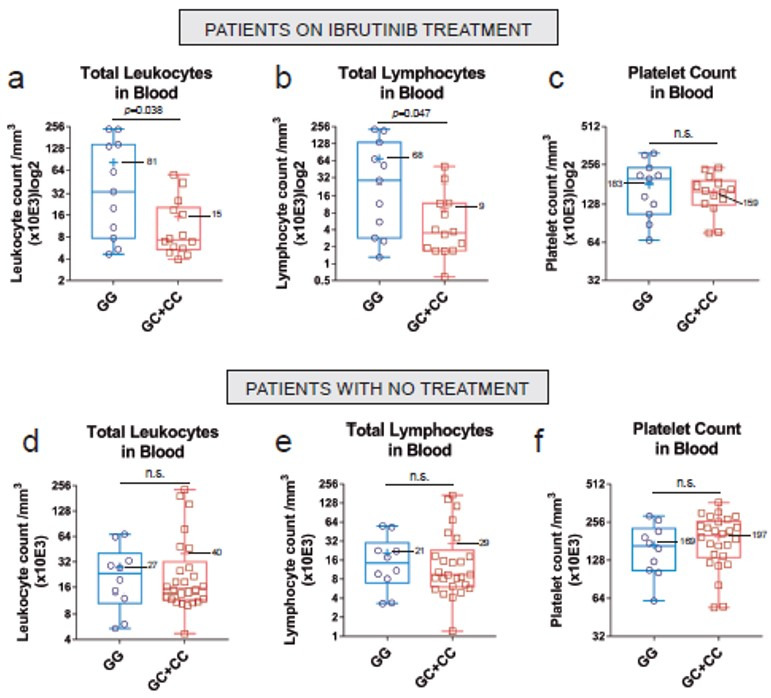
The presence of at least a C at rs8570 is associated with a better treatment response. (**a**–**c**) Box and whiskers plots showing all points (*n* = 26), median and mean values of the total leukocyte, lymphocyte and platelet count, respectively, in the blood of Ibrutinib-treated patients harboring at least one C allele (GC+CC) or none (G). Mean time under treatment = 23 months; median time under treatment = 13 months. (**d**–**f**) Box and whiskers plots showing all points (*n* = 36), median and mean values of the total leukocyte, lymphocyte and platelet count, respectively, in the blood of untreated patients as in (**a**–**c**). Statistical significance was assessed using a two-tailed unpaired t-test with Welch’s correction. n.s: not significant.

## Data Availability

All data are provided in the Appendix A.

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
