# Peer review of "An Optimized Single Nucleotide Polymorphism-Based Detection Method Suggests That Allelic Variants in the 3’ Untranslated Region of *RRAS2* Correlate with Treatment Response in Chronic Lymphocytic Leukemia Patients"

_cancers, 2023, doi:10.3390/cancers15030644_

Round 1

Reviewer 1 Report

In the manuscript the authors described a new and fast method to detect RRAS2 SNP. They also found in a small CLL populations changes in the expression of SNP in patients treated with ibrutinib. This is a descriptive study of a method to detect SNP of RRAS2 in CLL patients. The observed phenomenon in ibrutnib patients does not provide any mechanistic insight or explanations. in addition, the results have to be reproduced and expanded to a higher number of patients to confirm any clinical or predictive vallue of the findings.

Author Response

We agree with the Reviewer on that the low number of Ibrutinib-treated patients analyzed does not allow us to be conclusive. Therefore, following his/her recommendations and those of the Editor, we have turned down conclusive statements and, in addition, presented the Figure 3-5 data as a "proof-of-concept" of the PCR method. 

The effect of Ibrutinib on RRAS2 expression is puzzling. We do not know the possible mechanisms, although we have shown in our previous publication (ref. 19, Hortal et al) that downregulation of RRAS2 in a CLL cell line results in diminished phosphorylation of Btk. So, it seems that there is an interplay between BCR, Btk and RRAS2. Our preliminary finding shown in the current paper open an new and interesting perspective that is definitively worth to follow in coming works.

Reviewer 2 Report

In this succinct manuscript, Hortal et al., present a single-qPCR technique to optimize the detection of SNPs variants for the gene RRAS2, which has been connected with chronic lymphocytic leukemia. The manuscript clearly might be divided into two separated (although related) parts: the single-qPCR technique to improve the detection of SNPs in CLL cells, and the potential connection between ibrutinib and RRAS2 expression. In my opinion, while the first part suggests a methodology advance, the second part looks more preliminary based on the small number of enrolled patients. For instance, it is probable that a bigger n might indicate significant changes in Figure 3. Overall, the work is well-presented and the conclusions are supported by the results presented. The first part is more related to methodology, but is of interest for potential applicability in clinics and fits with the scope of the special issue. The second part is very preliminary but, as a proof of concept, I have no further comments on it.

As minor comments: There are a few redundancies in the text (ie Previously, the previous, line 186: to that that, Figure legend 1, line 227, etc.) and some other figures lack further details to understand about patients clustering, etc. (ie. Figure 2)

Author Response

We agree with the Reviewer on that the small number of Ibrutinib-treated patients analyzed so far does not allow us to be conclusive. We have now turned down the statements and indeed proposed Figures 3-5 to be a proof-of-concept.

Regarding Figure 2, we have changed entirely and used a new plotting strategy that helps to group patients into GG, GC, and CC genotypes very clearly.